# Biosynthesis of Zinc Oxide Nanoparticles from *Acacia nilotica* (L.) Extract to Overcome Carbapenem-Resistant *Klebsiella Pneumoniae*

**DOI:** 10.3390/molecules26071919

**Published:** 2021-03-29

**Authors:** Elsayim Rasha, AlOthman Monerah, Alkhulaifi Manal, Ali Rehab, Doud Mohammed, Elnagar Doaa

**Affiliations:** 1Department of Botany and Microbiology, College of Science, King Saud University, Riyadh 11451, Saudi Arabia; malothman@ksu.edu.sa; 2Department of Drug and Toxicology, College of Pharmacy, King Saud University, Riyadh 11451, Saudi Arabia; reali@ksu.edu.sa; 3Department of Microbiology, Prince Mohammed bin Abdulaziz Hospital-National Guard Health Affairs, Medina 41311, Saudi Arabia; doudmo@ngha.med.sa; 4Department of Zoology, College of Science, King Saud University, Riyadh 11451, Saudi Arabia; delnaggar.c@ksu.edu; 5Zoology Department, Faculty of Women for Arts, Science and Education, Ain Shams University, Cairo 11511, Egypt

**Keywords:** zinc oxide nanoparticle, wound healing, *Acacia nilotica*, carbapenem-resistant *Klebsiella pneumoniae* (KPC), green synthesis, antimicrobial resistance

## Abstract

Recently, concerns have been raised globally about antimicrobial resistance, the prevalence of which has increased significantly. Carbapenem-resistant *Klebsiella pneumoniae* (KPC) is considered one of the most common resistant bacteria, which has spread to ICUs in Saudi Arabia. This study was established to investigate the antibacterial activity of biosynthesized zinc oxide nanoparticles (ZnO-NPs) against KPC in vitro and in vivo. In this study, we used the aqueous extract of *Acacia nilotica* (L.) fruits to mediate the synthesis of ZnO-NPs. The nanoparticles produced were characterized by UV-vis spectroscopy, zetasizer and zeta potential analyses, X-ray diffraction (XRD) spectroscopy, Fourier transform infrared spectroscopy (FTIR), scanning electron microscopy (SEM), energy-dispersive X-ray spectroscopy (EDX), and transmission electron microscopy (TEM). The antimicrobial activity of ZnO-NPs against KPC was determined via the well diffusion method, and determining minimum inhibitory concentration (MIC) and minimum bactericidal concentration (MBC), the results showed low MIC and MBC when compared with the MIC and MBC of Imipenem and Meropenem antibiotics. The results of in vitro analysis were supported by the results upon applying ZnO-NP ointment to promote wound closure of rats, which showed better wound healing than the results with imipenem ointment. The biosynthesized ZnO-NPs showed good potential for use against bacteria due to their small size, applicability, and low toxicity to human cells.

## 1. Introduction

The increasing spread of multidrug-resistant bacteria in healthcare environments has left physicians with fewer treatment options, leading to costlier treatments [1]. Antimicrobial resistance (AMR) is considered a serious problem worldwide, increasing mortality and morbidity [2]. The leading cause of death in 2050 is predicted to be AMR, as Delo et al. asserted that “10 million people per year will die due to AMR infections and the mortality rate by AMR will be more than cancer.”

Over the last two decades, there has been a dramatic increase in the multidrug-resistant organisms’ CRE, which are generally difficult to treat, particularly among hospitalized patients. These organisms now constitute a high risk regarding antibiotic resistance in the United States, according to a report by the Center for Disease Control and Prevention (CDC) in 2013 [3]. Carbapenem antibiotics are used as a last resort to treat high-risk Gram-negative bacteria, a significant resistance source [4].

*Klebsiella pneumoniae* is an important pathogen that is involved in respiratory and urinary tract infections of hospitalized patients. It exists as part of the normal flora of the human respiratory tract and intestines and is isolated from the oropharyngeal cavity at a rate of 1–6% [5]. However, it also causes septicemia and iatrogenic infections. Moreover, it is associated with long-term hospitalized patients and those who have been in an intensive care unit (ICU) [5]. *K. pneumoniae* is resistant to ampicillin and carbenicillin and recently became resistant to cephalosporins, cotrimoxazole, aminoglycosides, and imipenem [5].

Currently, the trend for treating multidrug-resistant organisms has focused on the synthesis of metal nanoparticles by physical, chemical, or biological methods and their use as antimicrobial agents. Biological methods are considered the best option for synthesizing nanoparticles because they are more ecofriendly and safer than chemical methods, which are harmful to human cells [6].

The biosynthesis of nanoparticles from microbes and plant extracts has attracted researchers, attention because their small size, large surface area, orientation, and physical properties make them suitable to be used in medical sciences. Furthermore, they have a low cost and are not harmful to nature [7].

Zinc oxide (ZnO) is less commonly used as a nanomaterial in the medical field, especially against multidrug-resistant bacteria. The important features of ZnO nanomaterials are their low toxicity and biodegradability. Zn^2+^ is an essential trace element for adults and plays a critical role in different metabolism aspects. In the United States, 11.0 mg and 9.0 mg of Zn^2+^ per day is recommended for men and women, respectively. In terms of its chemical character, the ZnO surface is rich in -OH groups, which can be easily functionalized by different surface-decorating molecules [8].

One of the most important properties of ZnO is the possibility of producing particles that are small in size (less than a few hundred nanometers). This size helps in various applications because it is smaller than human cells [8]. Because of these properties ZnO have been used in the medical field for purposes such as biomedical imaging, treating skin rashes, and for sunscreens—as approved by the Food and Drug Administration (FDA) given their stability and inherent ability to absorb UV irradiation—as well as for drug and gene delivery in the treatment of cancer and autoimmune diseases and as antimicrobials [8].

The safety of ZnO-NPs has been studied by many researchers, who found that ZnO-NPs are nontoxic bioactive nanostructures that can be used as food packaging material and/or as a food preservative due to their properties of inhibiting biofilm formation and being safe for human cells [9]. The mod of action of ZnO-NPs on CRE appear by increasing the production of reactive oxygen species by the bacteria, which elevate membrane lipid peroxidation that causes membrane leak-age of reducing sugars, DNA, and proteins and reduce cell viability. These results indicate that ZnO-NPs could be developed as alternative drugs against CRE [10]. Thus, this study was established to investigate the antibacterial activity of biosynthesized ZnO-NPs against KPC in vitro and in vivo.

*Acacia nilotica* (L.) trees grow near rivers. Their seeds are traditionally used as an antiseptic or mixed with yogurt to treat dysentery, heal pustules in the skin, and treat sore throat and cough. These medicinal remedies are composed of flavonoids, phenolic compounds, and tannins from the seeds [11]. However, few studies have used *Acacia nilotica* to mediate nanomaterials synthesis, and this is the first study to use it as a reducing factor to produce ZnO-NPs.

Several studies have used aqueous extracts of *Acacia nilotica* as a bioreductant to obtain nanoparticles from silver-doped TiO_2_, iron nanoparticles (FeNPs), and gold nanoparticles and to assess their potential as antimicrobial and anticancer agents [12,13,14]. In this study, we used *Acacia nilotica* to mediate the synthesis of ZnO-NPs and subsequently tested it against KPC in vitro and in vivo.

## 2. Results

### 2.1. Characterization of Synthesized Zinc Oxide Nanoparticles

The optical characterization and confirmation of the ZnO-NPs synthesized using *Acacia nilotica* are presented in Figure 1A. The absorbance spectrum of ZnO-NPs showed a strong absorbance peak at 368 nm, which confirmed the formation of ZnO-NPs, as revealed using a UV-vis absorption spectrophotometer in the range of 200 to 800 nm. The purity and crystalline structure of ZnO-NPs were analyzed by XRD. The XRD spectrum of ZnO-NPs showed peaks at 2θ of 31.7°, 34.44°, 36.26°, 47.56°, 56.6° 62.89°, 66.39°, 67.97°, 69.1°, and 76.99° connected to surfaces at levels of (100), (002), (101), (102), (110), (103), (200), (112), (201), and (202), respectively (Figure 1B). The FTIR was used to identifi the functional groups that responsible for the synthesis of ZnO-NPs in both the sunthesized ZnO-NPs and the aquous extract of Acacia nilotica. The FTIR spectrum of ZnO-NPs after and before calcination and the aquoues extract of *acacia nilotica* showed an absorption band at 2922.61 cm^−1^, which is characteristic of the C-H medium stretching of an alkane group. The functional group C-N was detected in the absorption band 1201.13 cm^−1^ in all ZnO-NPs after and before calcination and the aquoues extract. ZnO fuctional group showed only in the absorption peak 434.31 cm^−1^ for ZnO-NPs after calcination. The rest of all absorption bands were presented in Figure 1C. These functional groups are present in various phytoconstituents like amines and alkaloids, cyanogenic glycosides, cyclitols, fatty acids and seed oils, fluoroacetate, gums, nonprotein amino acids, terpenes (including essential oils, diterpenes, phytosterol, triterpene genins, and saponins), hydrolysable tannins, flavonoids, and condensed tannins of plant extract, which are involved in the reduction and capping of synthesized ZnO-NPs [15].

The mean Z-average diameter (nm) of the ZnO-NPs and the size distribution showed promising results. As shown in Figure 2A, interestingly, the mean average size of the ZnO-NPs was observed to be about 94 nm. The size distribution profile showed two significant peaks with relative intensities of 96.4% and 3.6%. The polydispersity index was 0.263. On the other hand, the zeta potential value of ZnO-NPs was −50.4 mV (Figure 2B).

To confirm the presence of synthesized ZnO-NPs and their elemental composition, we used SEM imaging with EDX analysis. The SEM image characterizing the surface morphology of ZnO-NPs is presented in Figure 3A. It shows that some of the particles had an irregular spherical shape, and most of them presented a hexagonal shape with a smooth surface and were devoid of cracks. The size results matched those reported previously (6). The EDX analysis revealed the presence of zinc and oxygen components. The nanoparticles elemental analysis showed 83.43% zinc and 16.57% oxygen for weight% and 55.2% zinc and 44.8% oxygen (Table 1), Figure 3C,D.

TEM analysis was carried out on the ZOPs to obtain the size and shape of the nanoparticles. Morphological analysis of the NPs showed spherical, hexagonal, and rod-shaped structures, with some agglomerated large and small particles (Figure 3C). Regarding the size of the ZnO-NPs, SEM showed a nanoparticle size similar to that on TEM, ranging from 16 to 90 nm.

### 2.2. Antibacterial Activity

The most surprising aspect of our findings was in the zone of inhibition (ZOI) results, which showed the effects of the biosynthesized ZnO-NPs against KPC using an agar well diffusion method. The susceptibility test was conducted twice, with the results showing that all of the 20 tested KPC and the control (ATCC) were highly sensitive to ZnO-NPs, with a mean inhibition zone of 22.9 ± 1.96 mm at a concentration of 7.5 mg/mL. However, all tested bacteria were resistant to imipenem and meropenem antibiotics (Figure 4).

Zinc oxide nanoparticles showed high antibacterial potential against KPC and *Klebsiella pneumoniae* (ATCC 700603). The samples’ antimicrobial activities were evaluated using the ZOI, followed by the determination of the minimum inhibitory concentration (MIC) by the macrodilution method in culture broth. Figure 5A–C provide the experimental data of MIC and MBC for all tested bacteria; the mean score for MIC was 0.45 mg/mL and that for MBC was 1.14 mg/mL, indicating a significant difference between the average MIC and MBC (Table 2). In addition, the results showed that our ZnO-NPs were more potent than the commercial antibiotics in hospitals, such as imipenem and meropenem, as shown in Table 3.

### 2.3. SEM for Bacteria

The SEM analysis of KPC treated with ZnO-NPs revealed that its formulated nanoparticles caused more severe damage to the microbial cells, compared to untreated cells and those treated with impenem. Moreover, some of the bacterial cells were shown to change shape from rod-shaped to slightly coccus-shaped, while others decreased in size or showed multiple dents on the cell surface (Figure 3D–F). The SEM findings are consistent with those reported previously [16,17,18].

### 2.4. Effect of Zinc Oxide Nanoparticles on Improving Wound Healing

The assessment of wound healing was performed by observing the change of fresh wounds of rats and the degree of closure on days 3, 7, 11, and 14 after wounding (Figure 6). The infected and untreated control group (**G-1**) showed severe tissue inflammation with slight bleeding and purulence on the wound surface on all days. The uninfected and untreated control group (**G-2**) showed slight purulence on days three and seven, followed by a little bleeding from day 11 until day 14; the wound showed no significant healing or improvement. The infected and treated with imipenem ointment group (**G-3**), showed slight purulence on day three and considerable purulence on day seven; on days 11 and 14, there was an intermediate improvement, more significant than that in **G-1** and **G-2**. Finally, the group infected and treated with ZnO-NP ointment (**G-4**) showed a little purulence and bleeding on day three, which decreased on days 7 and 11. Then, on day 14, surprising results were observed, with the wound area being reduced almost entirely. Imaging results were confirmed by measuring the wound area on the same days (days 3, 7, 11, and 14). Analysis of the mean percentage of wound closure on day 14 after wounding showed that **G-1** and **G-2** had 63% and 64% healing, respectively, while **G-3** presented 54% healing and **G-4** exhibited 98% healing, as shown in Table 4. A histogram of Table 4 is shown in Figure 6.

### 2.5. Histopathology

Figure 7, Figure 8 and Figure 9 presented the hematoxylin and eosin (H and E)-stained sections of granulation/healing tissue from the four tested groups on days 3, 7, and 14. As shown in Figure 7A, the unwounded control skin showed a normal structure, with the surface showing a basket-weave keratinized layer (stratum cornea), followed by several layers of stratified squamous epithelia forming the epidermis layer that reached (302 µm). The epidermis layer was followed by a wide dermis layer consisting of connective tissue and collagenous fibers, in which hair follicles and blood vessels were embedded. Figure 7B, C shows tissue on day three after wounding. In **G-2**, the untreated and uninfected wounded skin showed removal of the epidermis layer with a wide scab forming, consisting mainly of collagenous fibers stained pale pink under which was a granulomatous leucocytic region (Figure 7B). As shown in Figure 7C, in **G-1** with untreated and infected wounded skin, there was a clear trend of increasing inflammation represented by removing the epidermis layer and the appearance of an abundant wide granulomatous leucocytic region interspersed with small and large hemorrhage spots stained cherry red. On day seven after wounding, **G-2** showed a scab consisting mainly of collagen stained pale pink interspersed with scattered inflammatory cells, under which was the dermis layer filled with collagen and scattered inflammatory cells (Figure 8A). Group 1 exhibited a wide and thick scab interspersed with hemorrhage batches under an abundance of inflammatory cells forming granulomas. The dermis layer was filled with scattered and aggregated inflammatory cells (Figure 8B). In **G-3**, there was a thin layer of scab with a wound edge at the skin’s surface, followed by a wide dermal area filled with a significant aggregation of inflammatory cells (Figure 8C).

In contrast, Group 4 showed a scab with granulomas followed by the dermis filled with scattered inflammation (Figure 8D). Finally, on day 14, after wounding, **G-2** indicated development of the wound represented by a thin scab prefaced with edema, under which was an abundance of granuloma (Figure 9A). In **G-1**, there was a thickened scab under an abundance of granuloma and hemorrhage (Figure 9B). In **G-3**, there was a scab followed by a regenerated layer of the epidermis and also regenerated dermis (Figure 9C). Group 4 showed improvement and skin regeneration manifested by a thickened regenerated epidermal layer covered with a keratinized layer (Figure 9D).

## 3. Discussion

The results of this study indicate that zinc oxide nanoparticles synthesized by the plant *Acacia nilotica* were stable for a long time, had a strong effect against resistant bacteria, and were safe for animal cells (showing low toxicity). Several characterizations of ZnO-NPs confirmed these findings and in vitro and in vivo experiments. The UV visualization results showed a characteristic band of light absorption for zinc oxide nanoparticles in the range of 360–380 nm. This is similar to the results reported by Zare et al., who showed that the absorption peak for synthesized ZnO nanoparticles was in the range of 360–376 nm [19]. Other similar results were reported by Padalia et al. and Safawo et al., who found that ZnO-NPs synthesized using *A. hydrophila* and *O. europea* extract showed an absorption peak 374 nm. The absorption maximum of ZnO-NPs synthesized using *C. abyssinica* occurred around 365 nm [6,20]. For zeta sizer and zeta potential, our findings appear to be consistent with those of Okpara et al. regarding the zeta potential (−38 mV) result, which was high. However, our findings disagree with their results regarding the nanoparticle size, with their result being more extensive than our NP size. This highlighted that our synthesized particles had little agglomeration and variation in size, findings that were more supported by Kim et al. They tested four ZnO-NPs synthesized by a physiochemical method, with their results showing good potential (26, 28, −39, and −42 mV) [21,22]. This study produced results that corroborate the findings of much of the work of Aldalbhi et al., who showed that the mean average size of NPs was about 94 nm [23]. Our nanoparticles results indicate that ZnO-NPs are stable due to the electrostatic repulsive force. The XRD findings were encouraging because they were comparable to those reported previously [6,20], but they are different from those of Aldalbahi et al. Our data showed a crystalline hexagonal structure of ZnO-NPs, similar to the result reported by Safawo et al [20,23]. These results indicate that the formed ZnO-NPs are pure and have an excellent crystalline structure due to their tight and strong diffraction peaks. The FTIR results showed that half of the plant extract peaks were present in ZnO-NPs before and after calcination, which supports the status of the plant extract as a reducing agent.

While most of the functional groups detected as shown in Figure 1C—such as C–H, C=O, C–C, N–H, N–O, C–N, OH, and ZnO—were similar to those obtained previously [10,24,25,26], the rest of the functional groups differed from our results because they used different types of plants. The average particle size of the ZnO-NPs, according to the SEM image, was around 48–88 nm, with a hexagonal shape similar to that reported previously [20,27,28]. The mean size of ZnO-NPs was 28 nm according to TEM images. However, the nanoparticles distribution histogram showed that the mean of ZnO-NPs is 28 nm. Such results were in agreement with previous reports [20,23], showing spherical, hexagonal, and rod-shaped structures, like our synthesized nanoparticles. The results of Kim et al. further support our results on nanoparticles size [21]. However, the current study found that the ZOI mean for each tested bacterium and the control ATCC was more than 19 mm, which means that the bacteria were susceptible to the ZnO-NPs according to the Clinical and Laboratory Standards Institute (CLSI) [29]. This study produced results that corroborate the findings of numerous previous studies on ZnO-NP activity against *Klebsiella pneumoniae* and beta lactamase-resistant bacteria. Many researchers tested the use of ZnO-NPs to overcome *Klebsiella pneumoniae* and their results agree with our own. For example, Yousef and Danial, Sharma et al., Jesline et al., and Farzana et al. obtained results of 25, 17, 16, and 27 mm, respectively [18,30,31].

The MIC result of using ZnO-NPs against KPC presented the strongest activity when compared with the results reported by Yousef et al. Their MIC result was 500 mg/mL against *Bacillus subtilis*, *Bacillus megaterium*, *Sarcina lutea*, *Klebsiella pneumoniae*, and *Proteus vulgaris* [30]. This high MIC result may be due to the fact that they were performing the chemical synthesis of ZnO-NPs. However, our nanoparticles were synthesized biologically, although our nanomaterial was dissolved in water. Nonetheless, these nanoparticles achieved high stability and potency as antibacterial, according to our MIC and MBC results. The morphological changes of KPC before and after treatment were evaluated by SEM imaging. The results showed transformations in cell size and shape due to the effect of ZnO-NPs; these changes led to the loss of the integrity of the membrane and the death of the cell [16]. The most interesting finding was obtained in vivo because it confirmed the potential of ZnO-NPs to overcome KPC. Specifically, according to the percentage of wound healing, we found that wound areas of **G-1** and **G-2** showed slight recovery via immune responses, while **G-3** showed intermediate healing because of the resistance of the tested bacteria to imipenem. However, we found that the wound closure result was the best in **G-4**. The wound healing of rat skin before and after treatment with imipenem and the green-synthesized ZnO-NPs was also evaluated by determining the recovery rate on soft tissue, as confirmed by H and E staining. From the histological results on days 3, 7, and 14 after wounding, the most striking result to emerge from the images was **G-4**. It showed improved wound healing and decreased bacterial inflammation compared with the antibiotic imipenem findings. These results corroborate the findings of many previous studies on infected wound closure and treatment with ZnO-NPs and support the result of Gao et al., who reported that ZnO-NPs are safe as approved by the FDA and suitable for the treatment of tissue damage which infected with a bacterial infection [32].

## 4. Materials and Methods

### 4.1. Synthesis of ZnO-NPs Using Aqueous Plant Extract

*Acacia nilotica* fruits were purchased from a local market, washed with distilled water (DW), and dried at room temperature. Twenty grams of *Acacia nilotica* fruits were grind with an electric grinder, mixed with 150 mL of sterile DW, and boiled in a stirrer heater at 80 °C for 2 h, the sample filtered through Whatman No. 1 filter paper. The aqueous extract was boiled until the stirrer heater reached 80 °C. After that, we added 5 gm of zinc nitrate obtained from Sigma-Aldrich Co. (Budapest, Hungary) and boiled the mixture for 2 h, causing the color to change to a deep brownish yellow [33]. After that step, we modified this method to create ZnO-NPs with right size and stability. The mixture of zinc nitrate and aqueous extract was kept at 60 °C for 24 h to obtain a dark brown spongy paste. Finally, we heated that spongy paste in a muffle furnace at 400 °C for 2 h to obtain a pure ZnO-NPs as a white powder and store it until use (Figure 10).

### 4.2. Characterization of Synthesized Zinc Oxide Nanoparticles

UV-vis confirmed the optical properties and formation of ZnO-NPs in the 200–800 nm range at a resolution of 10 nm. The optical properties of the nanoparticles were analyzed using UV-vis spectroscopy (UV-1800; Shimadzu UV Spectrophotometer, Kayoto, Japan). The particle size of the ZnO-NPs was determined by dynamic light scattering measurement, and zeta potential analysis was performed with a Malvern Zetasizer Nano series compact scattering spectrometer (Malvern Instruments Ltd., Malvern, UK). The histogram was developed by Zetasizer software (version 7.11) (Malveren Panalytical, Malveren, UK). Zeta potential was measured by the same instrument using a folded capillary cell. To understand the optical properties of the synthesized ZnO-NPs and to identify and determine the various modes of vibrations and the different functional groups present in the aqueous plant extract and fungi and ZnO-NPs, we used FTIR in the range of 400–4000 cm^−1^ (Parkin Elmer, Spectrum BX, Waltham, UK). To obtain information about the morphology and crystalline nature of the nanoparticles, we used XRD. We also measured the nanoparticles symmetry, size, and shape using an X-ray diffractometer (Brucker-Discover D8, CUK-alpha, Sangamon Ave, Gibson, USA) at a scan speed of 2deg/min in the range of 10–100°.

The size and composition of the ZnO-NPs were determined using a SEM (JEOL model, JSM-761OF, Tokyo, Japan) operated at an accelerating voltage of 10 kV with an EDX detector. TEM images were obtained using HRTEM (Jeol/JEM 1400, LaB6, Pleasanton, CA, USA) at an accelerated voltage of 80 kV, 150,000×, to determine the size and shape of the ZnO-NPs.

### 4.3. Determination of Antibacterial Activity of ZnO-NPs by Using Agar Well Diffusion Method, Minimum Inhibitory Concentration (MIC), and Minimum Bactericidal Concentration (MBC)

Twenty clinical isolates of KPC were isolated from wounds of ICU patients at Prince Mohammed Bin Abdul Aziz Hospital-Al Madinah. *Klebsiella pneumoniae* (ATCC 700603) was obtained from the College of Applied Medical Science, King Saud University. For the well plate agar diffusion method, the microbial cultures after overnight incubation on nutrient broth were adjusted to 0.5 McFarland turbidity standards and the suspension of each tested bacterium was streaked on a Mueller–Hinton agar (MHA) plate. About 1 mL of clinical isolates and ATCC were inoculated in Mueller and Hinton agar using a swab. A sterile cork borer was used to form wells (6 mm in diameter) on the agar plates. Then, 7.5 mg of ZnO-NPs was dissolved in 1 mL of deionized water to obtain a concentration of 7.5 mg/mL, followed by sonication and addition to each well in the culture plates that had already been inoculated with the test organisms [34]. The plates were then incubated at 37 °C for 24 h. After the incubation, the antimicrobial activity was determined by measuring the ZOI; this test was performed twice. Determination of the MIC was performed according to the CLSI using the broth macro dilution procedure [29]. The MBC was performed by obtaining 0.5 mL from MIC tubes that did not show any visible signs of growth and inoculated on sterile Mueller–Hinton agar by streaking. Then, the plates were incubated at 37 °C for 24 h. The result was read by recording the concentration at which no visible growth was seen, which was considered as the MBC [35].

### 4.4. Preparation of Bacterial Samples for Scanning Electron Microscopy (SEM)

SEM assessed the morphological changes of bacteria after ZnO-NP treatment. Briefly, 1 mL of KPC suspension (10^8^ CFU/mL) was used for mixing with 1 mL of ZnO-NPs to reach a final concentration of 0.5 mg/L, followed by cultivation overnight at 37 °C. Two controls (negative control with only KPC with medium and positive control with KPC treated with the antibiotic impenem) were prepared in salt-free lysogeny broth medium alone. After the incubation, the samples were washed with normal saline by centrifugation at 1500× *g* and then fixed with 2.5% glutaraldehyde at 4 °C for 2 h, followed by washing with phosphate buffer (pH = 7.2). The samples were post-fixed in 1% osmium tetroxide, followed by dehydration through an ascending ethanol series, critical point drying, and coating with Au–Pd (80:20) using a Polaron E5000 sputter coater, Quorum Technologies, Laughton, UK. The samples were checked at an accelerating voltage of 25 kV in FEI Quanta 250 using an SE detector [17,18].

### 4.5. Animal Preparation

Male Sprague–Dawley (SD) rats aged 12–14 weeks old were obtained from the Animal House of the College of Pharmacy, King Saud University, Riyadh, Saudi Arabia. The rats were housed in individual plastic cages lined with wood shavings, maintained on a 12:12 h dark-light cycle, and fed standard laboratory rat chow and filtered tap water (Liu et al., 2017). All animal experiments were approved by the Animal Care and Use Committee at King Saud University (Ethics Reference Number: KSU-SE-1978).

### 4.6. Wound Surgery and Induction of KPC Infection

Rats were anesthetized in accordance with the guidelines published by the University of California San Francisco, Office of Research Institutional Animal Care and Use. The intraperitoneal injection of ketamine/xylazine was performed at a dosage of 0.5 mL/100 g (100 mg ketamine/kg of body weight and 5 mg xylazine/kg of body weight) [36]. After we anesthetized the experimental rats, we removed their dorsal hair using a shaver, followed by washing with a lather and water. We created a wound of approximately 2 cm on each rats dorsal area with a sterile dermal biopsy punch on a special surgery table [17]. The total number of rats was 20, which were divided into four groups: **G-1** (positive control), **G-2** (negative control), **G-3** and **G-4** (17). KPC infection was induced by inoculating 20 μL of KPC bacterial suspension (CFU 10^8^) on the day of surgery (Day 0), which was then covered with Tegaderm and the rats were kept without treatment for 1 day to aid bacterial proliferation [17]. On a postoperative day (POD), we took swab samples from each wound to confirm that those wounds had been infected with KPC. On day 3, we started the treatment with ZnO-NPs ointment [37]. Measurement of wound area was carried out on PODs 3, 7, 11, and 14 (38) by estimation with a meter ruler. The percentage of wound healing was calculated according to the following equation:(1)$$\%\;\mathrm{of}\;\mathrm{WH}\;=\;\frac{\mathrm{WA}0-\mathrm{WAn}}{\mathrm{WA}0}\times 100$$
where WH is wound healing, WA0 is wound area on day 0, WAn is wound area on day n, and n = 3, 7, 11, and 14 days [38,39].

### 4.7. Preparation of Gel-Based Ointments

Ointment formulations were prepared at 5 mg/mL by adding equal volumes of polyethylene glycol (PEG) 400 and 2000 and ZnO-NPs and imipenem, followed by boiling them at 65 °C for 5 min [40].

### 4.8. Wound Healing and Re-Epithelialization

H and E was used to stain the tissues collected from the wound areas and morphological and histological assessments [41,42,43,44,45]. Tissue sections were checked under a light microscope (Nikon, Eclipse i80), and images were taken at different magnifications using a Nikon mounted digital camera (OXM 1200C; Nikon, Japan).

### 4.9. Statistical Analysis

Statistical analysis of the rate of wound recovery was performed using multivariate analysis of variance, and the statistical analysis of MIC, MBC, and ZOI was performed using paired samples t-test and ANOVA, with the results presented as mean ± SD with SPSS statistical software version 22 (SPSS Inc., Chicago, IL, USA).

## 5. Conclusions

The present study was designed to determine the characteristics and effects of biosynthesized ZnO-NPs to overcome KPC. According to the chemical characterization of the ZnO-NPs, the green synthesis method achieved nanoparticles with good shape, size, stability, and effective components against bacteria. UV-Vis showed a strong peak at 368 nm which matches the nanoparticles distribution histogram results. The mean size of the nanoparticles was 28 nm. The XRD results confirm the purity and the crystalline form of ZnO-NPs. All functional groups which detected in the synthesized ZnO-NPs and the aqueous extract of acacia nilotica were analyzed by FTIR. O-H group and C-H group were found in both ZnO-NPs and the aquoues extract of the plant; these functional groups were responsible of the ZnO-NPs synthesis. ZnO-NPs at 0.45 mg\ml concentration were inhibiting the growth of bacteria; the MBC of ZnO-NPs that could efficiently kill the bacteria was a mean of 1.14 mg/mL. With regard to wound healing, investigation of the ZnO-NPs showed that they were effective for wound healing, suggesting their capacity to act as an effective antimicrobial tissue adhesive.

## Figures and Tables

**Figure 1 molecules-26-01919-f001:**
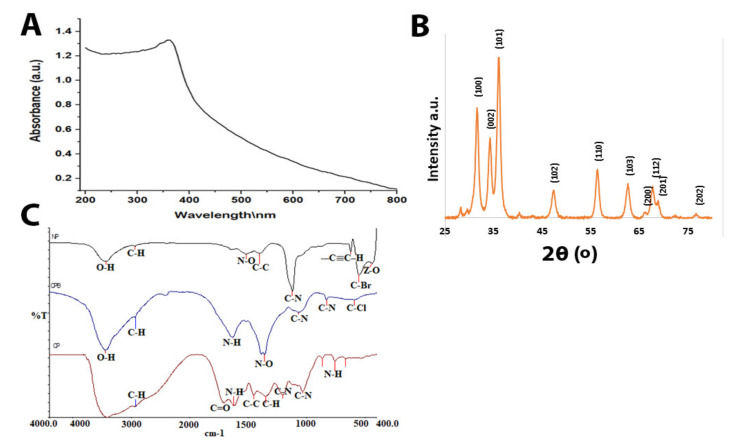
(**A**) UV–visible spectrum of ZnO-NPs. (**B**) XRD of ZnO-NPs. (**C**) FTIR of ZnO-NPs (NP = the formed ZnO-NPs after calcination, CFB = ZnO-NPs before calcination and CP = the aqueous extract of *Acacia nilotica*.

**Figure 2 molecules-26-01919-f002:**
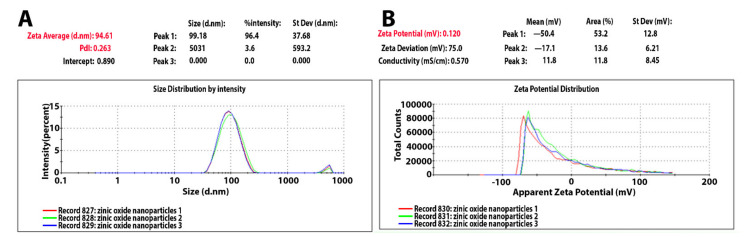
(**A**) Zetasizer of ZnO-NPs. (**B**) Zeta potential of ZnO-NPs.

**Figure 3 molecules-26-01919-f003:**
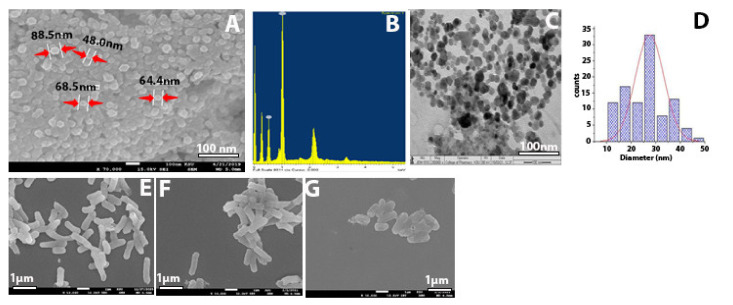
(**A**) SEM of ZnO-NPs. (**B**) EDX results of ZnO-NPs. (**C**) TEM of ZnO-NPs. (**D**) SEM of bacterial control. (**E**) KPC treated with imipenem at 500 mg/mL. (**F**) KPC treated with ZnO-NPs. (**G**) KPC treated with ZnO-NPs.

**Figure 4 molecules-26-01919-f004:**
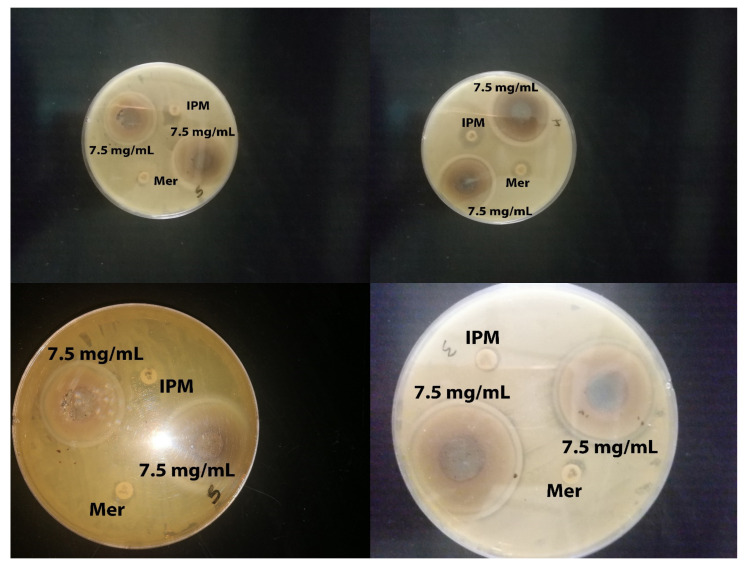
Zone of inhibition of ZnO-NPs against KPC at concentrations of 7.5 mg/mL. Controls: imipenem and meropenem.

**Figure 5 molecules-26-01919-f005:**
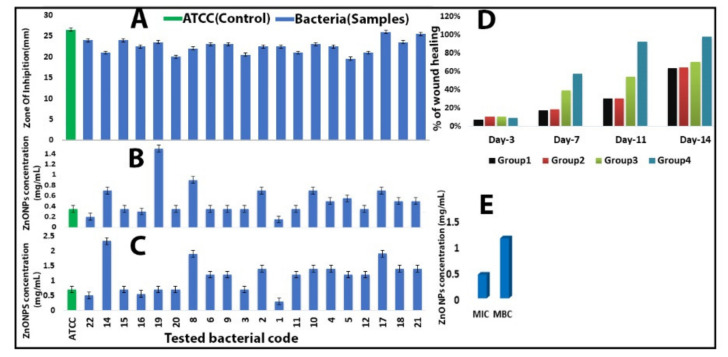
(**A**) Zone of inhibition (mm) of ZnO-NPs against *Klebsiella pneumoniae* (ATCC 700603) and tested bacteria. (**B**) MIC of ZnO-NPs against KPC and *Klebsiella pneumoniae* (ATCC 700603). (**C**) MBC of ZnO-NPs against KPC and *Klebsiella pneumoniae* (ATCC 700603). (**D**) The percentage of mean wound recovery in wound area within 14 days of wounding in Group 1 (infected and untreated control), Group 2 (infected and untreated control), Group 3 (infected and treated with imipenem), and Group 4 (infected and treated with ZnO-NPs). (**E**) Comparison between MIC and MBC.

**Figure 6 molecules-26-01919-f006:**
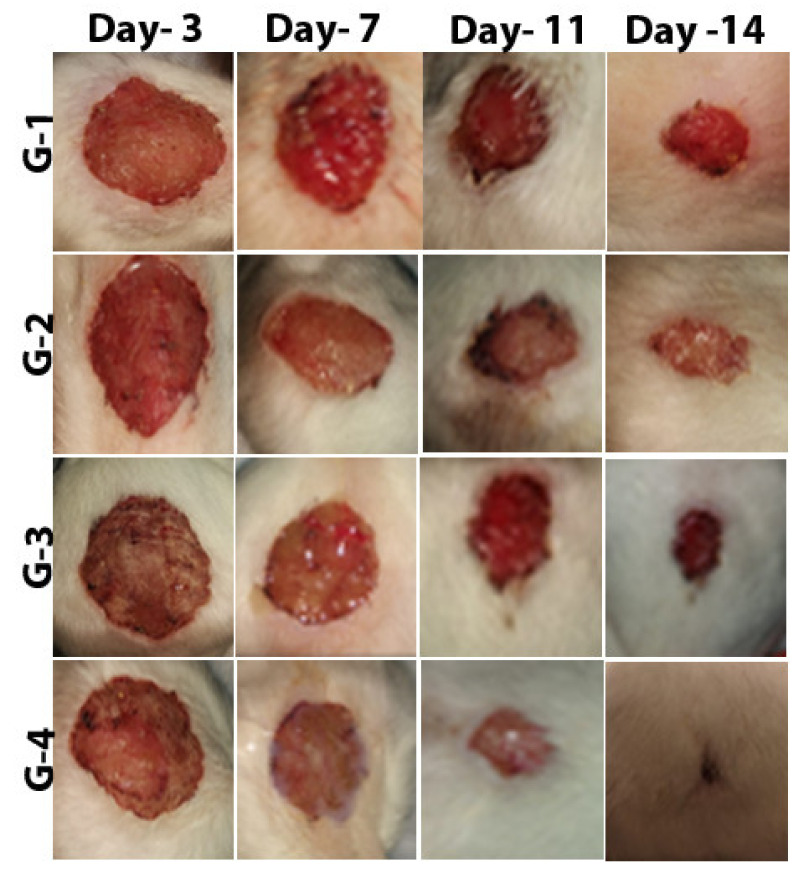
The stages of wound recovery in wound area within 14 days of wounding in Group **G-1** (infected and untreated control), Group **G-2** (uninfected and untreated control), Group **G-3** (infected and treated with imipenem), and Group **G-4** (infected and treated with ZnO-NPs).

**Figure 7 molecules-26-01919-f007:**
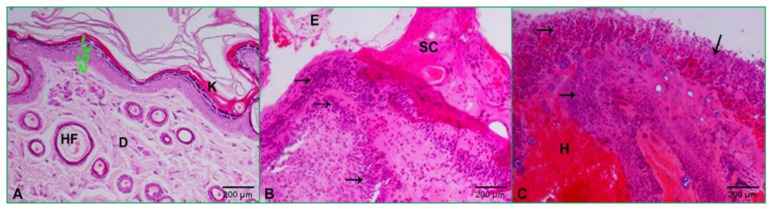
Photomicrographs of skin at day three after wounding: (**A**) control skin showing normal structure with several layers of stratified squamous epithelia (double-headed green arrow = 302 µm), keratinized layer (**K**), dermis (**D**), the hair follicle (**HF**); (**B**) untreated and uninfected wounded skin displaying edema (**E**), scab (**SC**), and granuloma (black arrows); (**C**) untreated and infected wound revealing wide granulomatous area (black arrows) and hemorrhage (**H**) (HE: 200×).

**Figure 8 molecules-26-01919-f008:**
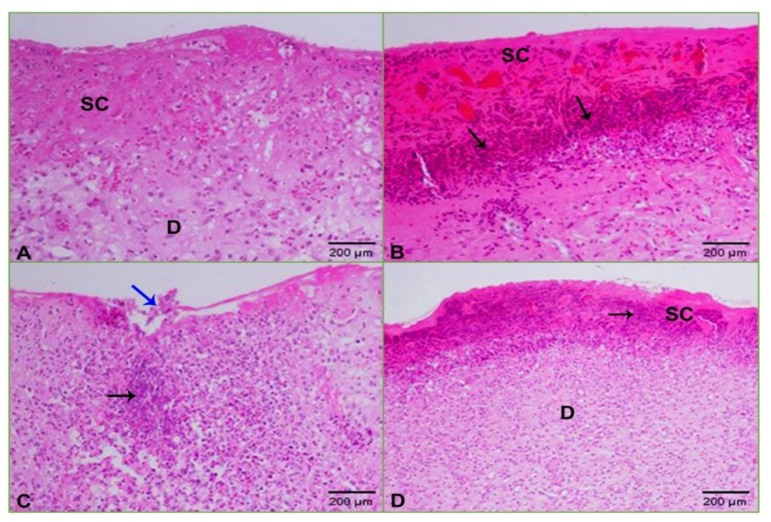
Photomicrographs of skin at day seven after wounding: (**A**) uninfected and untreated wounded skin showing a wide scab (**SC**) and scattered inflammatory cells in the dermis (**D**); (**B**) infected and untreated wounded skin revealing a scab (**SC**) interspersed with hemorrhage and granuloma (arrows); (**C**) infected and wounded skin treated with ZnO-NPs displaying a thin layer of scab with a wound edge (blue arrow) under which was a wide dermis area filled with inflammation (black arrow); (**D**) infected and wounded skin treated with imipenem exhibiting a scab (SC) interspersed with granuloma (arrow) and wide dermis (**D**) area filled with inflammatory cells (HE: 200×).

**Figure 9 molecules-26-01919-f009:**
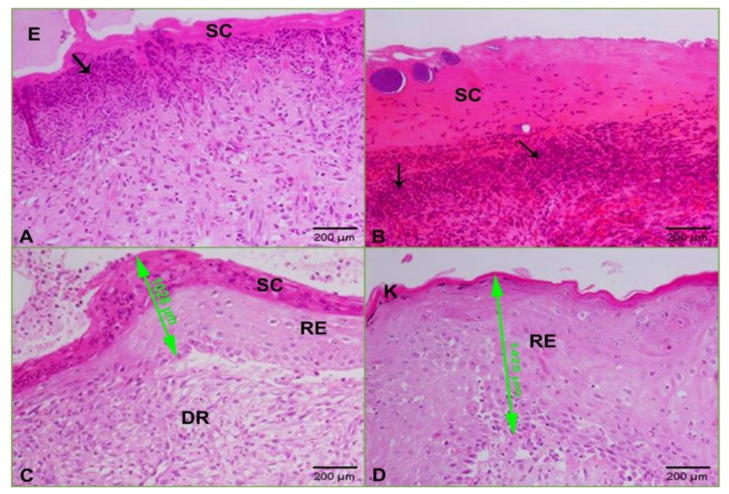
Photomicrographs of skin at day 14 after wounding: (**A**) uninfected and untreated wounded skin showing a scab (**SC**), granulomas (arrow), and edema (**E**); (**B**) infected and untreated wounded skin revealing a wide scab (**SC**) under which were granuloma (arrows) and hemorrhage; (**C**) infected and wounded skin treated with ZnO-NPs displaying a scab (**SC**) under which was regenerated epithelia (green arrow) of 1028 µm followed by regenerated dermis (**DR**); (**D**) infected and wounded skin treated with imipenem showing regenerated skin covered with a keratinized layer (**K**) of the thickened regenerated epidermis of 1425 µm (green arrow) (HE: 200×).

**Figure 10 molecules-26-01919-f010:**
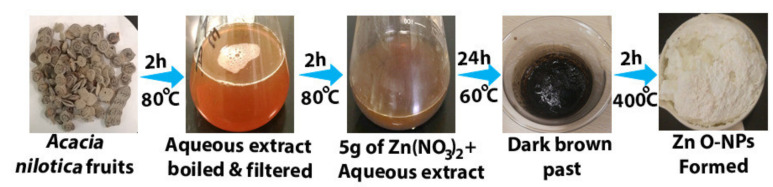
ZnO-NPs synthesized steps by using *Acacia nilotica* fruits.

**Table 1 molecules-26-01919-t001:** Zinc and oxygen elements analyzed by weight % and atom %.

Element Line	Element Wt.%	Atom%
**O**	16.57	44.80
**Zn**	83.43	55.20
**Total**	100.00	100.00

**Table 2 molecules-26-01919-t002:** Comparison between MIC and MBC.

Mean ± Std. Deviation	
**MIC**	0.45 ± 20
**MBC**	1.14 ± 50

**Table 3 molecules-26-01919-t003:** MIC of imipenem and meropenem antibiotics against *Klebsiella pneumoniae* (ATCC 700603) and tested bacteria.

Bacterial Code	Meropenem	Imipenem
**ATCC**	>0.25	>1
**22**	>16	>2
**14**	>4	>4
**15**	>16	>4
**16**	>16	>4
**19**	>16	>4
**20**	>16	>4
**8**	>16	>4
**6**	>16	>4
**9**	>16	>4
**3**	>16	>16
**2**	>16	>4
**1**	>16	>4
**11**	>4	>4
**10**	>16	>4
**4**	>4	>2
**5**	>4	>4
**12**	>16	>4
**17**	>16	>4
**18**	>16	>4
**21**	>16	>4

**Table 4 molecules-26-01919-t004:** The percentage of mean wound recovery in wound area within 14 days of wounding in Group 1 (infected and untreated control), Group 2 (infected and untreated control), Group 3 (infected and treated with imipenem), and Group 4 (infected and treated with ZnO-NPs).

Groups	Days				
	0 Day	Day-3	Day-7	Day-11	Day-14
**G-1**	2cm	7 ± 2.739	17 ± 2.739	30 ± 5.000	63 ± 5.701
**G-2**	2cm	10 ± 5.000	18 ± 6.708	30 ± 7.071	64 ± 9.618
**G-3**	2cm	10 ± 3.536	39 ± 8.944	54 ± 4.183	70 ± 6.124
**G-4**	2cm	9 ± 4.183	57 ± 4.472	92 ± 5.701	98 ± 2.739

## Data Availability

Not applicable.

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
