# Peer review of "Biosynthesis of Zinc Oxide Nanoparticles from Acacia nilotica (L.) Extract to Overcome Carbapenem-Resistant Klebsiella Pneumoniae"

_molecules, 2021, doi:10.3390/molecules26071919_

Round 1
Reviewer 1 Report
This is an interesting work, but the writing of the article is too long, making the readers tasteless and incomprehensible. It is recommended to shorten the length of the article and be simple and concise. The writing of the article is too poor, and the description is not clear in many places. There are too many issues. First of all, I feel very confused about the structure. Many details of the structural representation are not clearly described. Many details of the characterization are completely ignored. The quality of the pictures in the article is also very low. In short, if this article is to be published, there are still many parts that need to be carefully improved. I recommend publication of this work after they address properly my comments.
- There are some grammatical errors in the text, and figure 8 and the caption do not correspond accordingly, please correct them carefully.
- The explanation of some figures is not clear enough, and the quality of all the figures in this manuscript need to be improved with high-resolution images, also the SEM images should be added much clearer scale bar in order to be published. There are many figures that can be merged, for example, I strongly suggest to combine Figure 6 to 8.
- Please provide a Gaussian fitting histogram of the nanoparticle distribution, instead of directly marking the size of the nanoparticle on the TEM or SEM image. This is an article to be published, not a draft.
- The authors should provide the reference PDF card of ZnO in figure 4. The result of EDX test should be atomic ratio instead of mass ratio. The atomic ratio to mass ratio is clearer, please correct it.
In short, these comments are only comments from the perspective of the overall structure, and the author should first properly resolve these issues. After this, please return to the reviewer for the second round of comments for the revision of the content of this article.
Author Response
Dear,
We revised all your comments as you requested.
for the XRD reference pdf please check it in the email I sent to the Assistant editor and certificate of editing of the manuscript.

Reviewer 2 Report
This article presents the characteristics and effects of biosynthesized ZnO-NPs to overcome KPC. The complexes are characterized from a structural point of view. All conclusions are supported by the experimental data and the experiments were well conducted. I find the result of this work meaningful. This work will be suitable for publication after addressing the following point:
- Re-draft the introduction part with only the most important content associated with the work.
- Page 4. “As shown in Figures 2 and 3, interestingly, the mean average size of the ZnO-NPs was observed to be about 94 nm.” Since Figure 3 does not describe the size distribution, the sentence should be changed.
- Page 5. “The purity and crystalline structure of ZOPs were analyzed by X-ray diffraction (XRD) spectroscopy.” How do you confirmed the purity and crystalline structure since only one XRD of ZnO-NPs synthesized from Acacia nilotica with any comparison?
Author Response
Dear,
We revised all your comments as you requested.

Round 2
Reviewer 1 Report
Although some of the responses deserve a debate, most of the comments are properly addressed. The manuscript could be accepted.